# Plant Aquaporin Gating Is Reversed by Phosphorylation on Intracellular Loop D—Evidence from Molecular Dynamics Simulations

**DOI:** 10.3390/ijms241813798

**Published:** 2023-09-07

**Authors:** Robin Mom, Stéphane Réty, Vincent Mocquet, Daniel Auguin

**Affiliations:** 1Laboratoire de Biologie et Modélisation de la Cellule, ENS de Lyon, University Claude Bernard, CNRS UMR 5239, INSERM U1210, 46 Allée d’Italie Site Jacques Monod, F-69007 Lyon, France; stephane.rety@ens-lyon.fr (S.R.);; 2Research Group on VestibularPathophysiology, CNRS, Unit GDR2074, F-13331 Marseille, France; 3Laboratoire de Biologie des Ligneux et des Grandes Cultures, Université d’Orléans, UPRES EA 1207, INRA-USC1328, F-45067 Orléans, France

**Keywords:** aquaporin, PIP, gating, phosphorylation, in silico analysis

## Abstract

Aquaporins (AQPs) constitute a wide and ancient protein family of transmembrane channels dedicated to the regulation of water exchange across biological membranes. In plants, higher numbers of AQP homologues have been conserved compared to other kingdoms of life such as in animals or in bacteria. As an illustration of this plant-specific functional diversity, plasma membrane intrinsic proteins (PIPs, i.e., a subfamily of plant AQPs) possess a long intracellular loop D, which can gate the channel by changing conformation as a function of the cellular environment. However, even though the closure of the AQP by loop D conformational changes is well described, the opening of the channel, on the other hand, is still misunderstood. Several studies have pointed to phosphorylation events as the trigger for the transition from closed- to open-channel states. Nonetheless, no clear answer has been obtained yet. Hence, in order to gain a more complete grasp of plant AQP regulation through this intracellular loop D gating, we investigated the opening of the channel in silico through molecular dynamics simulations of the crystallographic structure of *Spinacia oleracea* PIP2;1 (*So*PIP2;1). Through this technique, we addressed the mechanistic details of these conformational changes, which eventually allowed us to propose a molecular mechanism for PIP functional regulation by loop D phosphorylation. More precisely, our results highlight the phosphorylation of loop D serine 188 as a trigger of *So*PIP2;1 water channel opening. Finally, we discuss the significance of this result for the study of plant AQP functional diversity.

## 1. Introduction

Aquaporins (AQPs) are transmembrane water channels found in almost all living organisms [1]. They constitute a wide protein family that is especially diverse in plants, where it is common to find dozens of homologues [2,3]. Even though a part of the underlying functional diversity remains misunderstood to this day, this high number of dedicated water channels in plants probably stems from their very high phenotypic plasticity associated with the necessity for them to cope with an ever-changing water environment [4].

Decades of investigations have resulted in a better understanding of the complexity of water flux regulation through AQPs by shedding light upon all the entangled regulatory mechanisms [5]. Indeed, membrane water permeability can be regulated through AQP transcript level modulation [6] and post-translational modifications [7]. These two levels of regulation are directly linked to vesicular trafficking [8], heteromerization [9], or gating of the AQP [10]. In fact, this wide range of membrane water permeability dynamic regulations is thought to be essential to maintain plant water homeostasis and is integrated into the functional diversity of the AQP family [5,11].

AQPs are naturally found as tetrameric assemblies [12]. Each subunit constitutes a functional water channel made by six transmembrane alpha helices and two small hemi helices responsible for the typical electrostatic profile of the conducting pore (Figure 1) [12]. This profile is characterized by single-file water channeling and proton exclusion [13,14,15]. On the extracellular and intracellular surfaces of the tetramer, flexible loops allow for more specific regulations such as the intracellular-pH-associated gating mechanism described in plant AQPs [16].

Plant AQP gating involves structural changes in the intracellular loop D from an “open” to a “closed” conformation and has been thoroughly described by Susanna Törnroth-Horsefield and collaborators [16] (Figure 1C). This regulation mechanism has been demonstrated through the resolution of two experimental structures of spinach AQP *So*PIP2;1. In the “closed” conformation, the loop D is positioned in such a way that water molecule passage is sterically halted. Indeed, the hydrophobic sidechains of leucine 197 (L197), histidine 99 (H99), valine 104 (V104), and leucine 108 (L108) generate a hydrophobic occlusion, narrowing the diameter of the pore to 0.8 Angströms. This closed conformation is stabilized by the establishment of salt bonds between the charged sidechains of loop D arginine 190 (R190), aspartate 191 (D191), and residues of the N-terminal region (N-ter) such as lysine 33 (K33). A histidine of this same intracellular loop D, histidine 193 (H193), is supposed to strengthen these interactions and to orientate the loop toward the “closed” conformation upon its protonation. Indeed, the additional positive charge would allow salt bridge formation with N-ter aspartate 28 (D28) along with the reorganization of the hydrogen bond network with the loop B [16,17]. It is also hypothesized that the “closed” conformation of the loop D would be further stabilized by the fixation of a divalent cation by the N-terminal extremity (N-ter) glutamate 31 (E31) [16,17]. However, even though the “closed” state of plant AQPs is described in detail, data on the opening of the channel are more scarce, in particular because of the low resolution of the intracellular loop D in the “open” conformation [16]. From this initial study, a few molecular dynamics simulations and the conservation of two phosphorylation sites among plant AQPs formed the grounds for the hypothesis that serine 115 (S115) and serine 274 (S274) phosphorylation could trigger loop D opening [16]. Indeed, protein phosphorylation is well known for its role in cell signals transduction and has already been associated with the gating of other channels such as chloride ion channel CFTR (cystic fibrosis transmembrane conductance regulator) [18]. Moreover, plant AQP modulation by phosphorylation is already well described. PIP in particular has already been involved in the control of sub-cellular localization and in gating [8,19]. Another work explored the effect of intracellular phosphorylation of serine 115, serine 188, and serine 274 through the characterization of water permeability of *Xenopus laevis* oocytes expressing *So*PIP2;1 with the serine of interest replaced by a glutamate (S115E, S188E, and S274E, respectively) [20]. The authors found no significant increase in water permeability for the S115E and S274E mutants, which were also crystallized in a closed conformational state. These results questioned the relevance of these two phosphorylation marks in *So*PIP2;1 gating regulation. However, in the same work, the S188E mutant was associated with significantly higher water permeability even though the authors could not manage to crystallize the corresponding open conformation [20]. Moreover, other molecular dynamics studies, issued from longer trajectories of 100 nanoseconds of length, could not capture any significant impact of S115 or S274 phosphorylation upon *So*PIP2;1 water permeability, contrasting again with the pioneering study of Susanna Törnroth-Horsefield and collaborators [21,22]. In order to further test the relevance of phosphorylation for the regulation of plant AQP gating, we investigated the loop D serine 188 phosphorylation mark through molecular dynamics approaches. By performing simulations exceeding one microsecond in length, we managed to capture the whole dynamic process of loop D conformational changes from an initial “closed” state toward a fully functional “open” state. We then proposed a detailed description of the underlying molecular mechanism. Finally, we discuss the relevance of our results and the physiological implications of such regulation in regards with the intrinsic diversity of plant AQPs.

## 2. Results

### 2.1. Opening of Loop D by Phosphorylation of Serine 188

In order to assess the phosphorylation of serine 188 (S188Phos) as an open state initiator, we simulated a tetrameric assembly of spinach SoPIP2;1 in an initial closed conformation extracted from an experimental protein structures database (pdb: 1z98 [16]) with serine 188 in a phosphorylated state in each subunit.

We observed the opening of the channel around 100 nanoseconds after the beginning of the simulation, as shown by the number of water molecules crossing the whole conducting pore (Figure 2A). Along the whole trajectory, all the subunits allowed for water to cross the membrane, however, this was in a very disparate way. Indeed, one subunit in particular, chain D, became highly functional as indicated by the stiff slope of cumulative water permeations curve (Figure 2A). Interestingly, we can see that this functional change is associated with a net widening of the intra-cellular vestibule, estimated through the mean minimal distance between backbone alpha carbons of loop D and N terminal residues (Figure 2B). This distance culminates at time t = 871 nanoseconds, which also corresponds to a very high water passage (Figure 2A,B). A schematic representation of the associated conformation is displayed in Figure 2C.

Because chain D displayed the most contrasted phenotype with the initial closed conformation, we decided to focus the following analysis on this subunit. To estimate the impact of the phosphorylation of serine 188, we used the first 200 nanoseconds of chain D trajectory as the “closed” control condition and the 200 nanoseconds portion of trajectory between t_start_ = 700 nanoseconds and t_end_ = 900 nanoseconds as the “open” condition (Figure 2B). This choice was based on both the number of water molecules crossing the whole transmembrane section of the channel and the mean minimal distance between the loop D and N terminal residues backbone (Figure 2A,B). The free energy profiles of water molecules inside the pore clearly illustrate the conformational change in loop D through the disappearing of the closed-state-associated free energy barrier in the “open” condition (i.e. at the end of the trajectory, Figure 2D). This free energy barrier is the highest of the pores, even though the “closed” condition also displays two other free energy barriers higher than the “open” condition: in the NPA region (where small helices HB and HE meet at the center of the channel; Figure 1) and at the aromatic/arginine constriction (in the extra-cellular part of the channel; Figure 1) (Figure 2D). To compare the two conditions, two different permeability indicators were used: the traditional permeability coefficient (pf) and the more straightforward number of water molecules crossing the whole conducting pore (Figure 2D). In both cases, the very small associated *p* values indicated the very significant impact of the phosphorylation on the channel water permeability (Figure 2D).

To better understand the molecular mechanism at stake, we focused on the residues mainly interacting with the phosphorylated serine 188 in chain D: arginine 187 and arginine 190 (Figure 3 and Appendix A). It appeared that the opening of the channel was precisely correlated with the establishment of a salt bridge between arginine 190 and the phosphoryl group (Figure 3A,B and Appendix A). This salt bridge was formed after a first one between arginine 187 and the phosphoryl group (Figure 3B). Hence, the opening of the channel happened as follows: the phosphorylation of loop D serine 188 brought two negative charges at this location. Because of the charges attraction, the closest positive charge, the guanidinium group of the adjacent arginine 187, was forced to counteract one negative charge of the phosphoryl group. Through a similar mechanism, another arginine located on the same loop formed a second salt bridge with the phosphoryl group. These changes in arginine 187 and arginine 190 side chain conformations (both being part of the loop D residues) induced a new curvature of the loop backbone, breaking the initial interactions established with N-terminal residues and triggering the opening of the channel (Figure 3C and Appendix A).

### 2.2. Stabilization of the Open Conformation

To further investigate the stabilization of loop D in such an open conformation, we built a new atomic system. The subunit chain D in an open state was extracted from the previous trajectory and duplicated into a tetrameric fully open assembly and the C-terminal extensions were removed (see methods, Figure 4A). From the cumulative number of water permeations over the 100 nanoseconds simulated, we can see that the four subunits stay functional (Figure 4A). However, when we looked at the formation of the two salt bridges previously identified as crucial for the channel opening (i.e. S188Phos—R187 and S188Phos—R190), we observed a heterogeneous situation between the four chains (Figure 4B). Depending on the subunit, the two salt bridges seemed to be formed and broken more or less frequently.

They appeared most stable in chain C and less in chains B and chain D. However, the stabilization of these salt bridges did not correlate with the water permeability of the subunit (Figure 4A,B). This can be explained by the compensation of S188Phos—R187 or S188Phos—R190 salt bridges rupture by the formation of new ones with other charged residues of loop D: aspartate 184 (D184), lysine 186 (K186), and aspartate 191 (D191) (Figure 4C). Altogether, this open state of the tetramer appeared stable over time independently from putative C-terminal extensions interactions. This open conformation was maintained through the establishment of a salt bridges network between the charged residues of loop D within and between chains (Figure 4C). The root mean square fluctuation (rmsf) profile of the protein backbone clearly illustrates the loop D and, to a smaller extent, the C-terminal extremities, as the less stable parts of the aquaporin over the 100 nanoseconds of simulation (Figure 4C). This can be explained by the fact that the S188Phos—R187 and S188Phos—R190 salt bridges were not stable over time. This maintained loop D in a flexible open conformation in a dynamic equilibrium.

### 2.3. Dephosphorylation Triggers the Closing of Loop D

Finally, we investigated the effect of serine 188 dephosphorylation on the closing of SoPIP2;1 water pore. In order to do so, four conditions were compared over a 500 nanoseconds timescale (see methods, experimental setup 3, Figure 5): condition “open_S188Phos” corresponds to the previously mentioned tetrameric construction of the open conformation with serine 188 phosphorylated; “open” to the same tetrameric open construction but without the phosphorylation mark; “open_H193P” to the open tetrameric assembly without phosphorylation and with histidine 193 protonated; and, finally, “closed_S188Phos” to the closed state of the tetramer with serine 188 phosphorylated, i.e., to the first 500 nanoseconds of simulation of the first experimental setup (see methods and Figure 2). Condition “open_H193P” was simulated because the protonation of this histidine was previously involved in the closing of the channel [16,23]. Condition “closed_S188Phos” was used as a control condition for the closing of the channel. The phosphorylation of serine 188 maintained the tetramer in a stable functional state over this longer timescale, as illustrated by the number of water molecules crossing the whole conducting pore section (Figure 5A). Indeed, the permeability of the three other conditions were significantly smaller than in “open_S188Phos” and correlated with significantly smaller distances between N-ter and loop D backbone atoms (Figure 5B). However, we could observe closed states of the pore (indicated by a plateau phase on the cumulative counts of water permeations curves) in all four conditions (Figure 5C). To compare these closed loop D conformations together, 100 nanoseconds portions of trajectory corresponding to a closed state were used to generate average structures of the closed sub-units. These closed monomers were then aligned together with the crystallographic structure of the closed conformation (pdb 1z98) using PyMOL 2.4 software [24] (Figure 5D). Except for “closed_S188Phos”, we can see that the loop D is not in the same conformation as in the crystal structure (pdb 1z98). However, even though the loop is not stabilized through the same hydrogen bonds and salt bridges network as in the crystal structure, the four residues responsible for the formation of the hydrophobic occlusion of the channel are positioned at the same location. In particular, leucine 197 of loop D is maintained close to histidine 99, leucine 108, and valine 104 (Figure 5D), which is sufficient to close the pore as described in the initial study by Susanna Törnroth-Horsefield [16].

Figure 5E presents the multiple alignment of SoPIP2;1 with six other PIP sequences from Spinacia oleracea. It appears that the position corresponding to serine 188 in SoPIP2;1 can bear other amino acids such as a lysine, an asparagine or an isoleucine. These residues differ strongly in nature with serine. Indeed, arginine is positively charged, isoleucine is hydrophobic, and the three of them cannot be phosphorylated, therefore implicating functional differences in the regulation of loop D gating between SoPIP2;1 and these three homologues.

To conclude, in all conditions, loop D can reach closed conformations. When serine 188 is phosphorylated, however, (condition “open_S188Phos”) this conformational state is a rare event as the overall higher water permeability and N-ter—loop D distance (Figure 5A,B) indicates. The dephosphorylation of serine 188 is sufficient to shift this dynamic equilibrium toward a rather closed, less-functional tetramer (conditions “open” and “open_H193P”). Despite this pivotal role, serine 188 is not strictly conserved among spinach PIP, suggesting the existence of other loop D opening mechanisms. The additional protonation of histidine 193 (condition “open_H193P”) known for its role in the closure of loop D does not induce a more significant closing of the channel (Figure 5). Finally, we can observe that a stable closed state is never obtained as the structural alignment with the experimentally resolved closed conformation highlights (Figure 5D). This difference indicates that one or more other molecular partners are needed to lock the interaction of loop D with the N-terminal extremity.

## 3. Discussion

In the present study, we evaluated, through in silico molecular dynamics, simulations of the impact of the phosphorylation of intra-cellular loop D serine 188 of SoPIP2;1 on the water permeability of the channel. We observed conformational changes in loop D leading to a very significant increase in water permeability (Figure 2). In their study, Nyblom and collaborators also observed a significant increase in pf of the mutant mimicking this phosphorylation (S188E) of about three times the pf of the native SoPIP2;1 [20]. We observed a difference of the same order of magnitude with a pf in the “open” condition being two times higher than in the “closed” condition (Figure 2). The convergence of these two approaches, despite the very different methodology (heterologous expression in Xenopus oocytes versus in silico molecular dynamics simulations) and time scales (seconds versus nanoseconds) involved, supports this result and the reliability of this phosphorylation mark as a trigger of SoPIP2;1 opening. In this same study of Nyblom et al., the mutant mimicking the phosphorylation of serine 188 could not be crystallized despite the several attempts made indicating the loop D probably being in a similar unstructured state as in the open conformation structure [20]. In our simulations, we observed the open conformation of loop D stabilized in an equilibrium involving the dynamic formation and rupture of salt bridges between the phosphorylated serine 188 and other charged residues of loop D (Figure 3 and Figure 4). These results are, hence, in good accordance with Nyblom et al. [20]. In their study, Nyblom et al. also simulated the impact of serine 188 phosphorylation upon the conformation of loop D. In their fifty nanoseconds simulations, they observed changes in loop D conformation but did not analyze their impact on the permeability of the channel. In comparison, the much longer timescales of our simulations coupled with a functional analysis of the channel yielded a more complete picture of this phenomenon. In our simulations, the open state of the channel could be reached and stabilized in a tetrameric assembly without C-terminal extensions (Figure 3 and Figure 4). Contrary to the hypothesis of Nyblom et al. [20], this result does not indicate the interaction with C-ter as mandatory for the stabilization of loop D, even though we have observed such interactions in the trajectory with the full protein.

Among the seven spinach PIP amino acids sequences compared through multiple alignment, four displayed a serine at position 188, and the three other a lysine, an asparagine or an isoleucine (Figure 5E). These differences would probably impact the regulation of loop D PIP gating and could account for a tissue-specific functional regulation diversity in plants. This regulation could be a way to precisely monitor the water permeability of plant membranes, especially when coupled with the heteromerization described in plant AQPs [9,25]. However, it also suggests that there are other mechanisms allowing for the opening of the channel independent from the phosphorylation of serine 188. Two other phosphorylation marks (S115Phos and S274Phos) were studied by Nyblom et al.; however, they yielded no increase in water permeability [20]. Another serine, located in the N-terminal extremities close to the interaction region with loop D (serine 36 in SoPIP2;1) could also trigger an opening of the channel upon phosphorylation. This hypothesis is supported by the functional characterization of TgPIP2;2 mutants mimicking the phosphorylation of this other serine [26].

Starting from a fully open conformation, the removal of serine 188 phosphorylation mark was sufficient to trigger a significant shift in the dynamic equilibrium of loop D conformations from an open state toward a rather closed channel (Figure 5). Interestingly, the protonation of histidine 193, hypothesized as a potential trigger of the closure of loop D [16,23], did not allow for a more rapid closure of the pore. Hence, this suggests that the protonation of histidine 193 is involved in the stabilization of loop D in a closed conformation rather than in the transition from the open state toward the closed state of the channel. However, this protonation was not sufficient by itself to lock the loop D in a closed conformation as depicted by the structural alignment of simulated conformations with the crystallographic structure of the closed state of *So*PIP2;1 (Figure 5D). This, together with the presence of a divalent cation in the closed X-ray structural state [16,17], indicates this cation as the probable missing key allowing for the formation of a stable closed loop D conformation.

Finally, this work provides additional evidence in favor of the physiological relevance of serine 188 phosphorylation and new insights into the molecular mechanisms involved in the opening of the channel and the stabilization of loop D in such an open conformation. The phosphorylation of serine 188 still remains to be shown in vivo. However, together with this serine being situated within a protein kinase C consensus phosphorylation site, the mechanistic hypothesis suggested by the present work further strengthens the significance of serine 188 as a key residue in the regulation of plant AQPs water permeability. The removal of this phosphoryl group was shown to be enough to trigger the closure of the pore in our simulations. However, more research is needed to better understand the relevance of histidine 193 protonation state in this mechanism as well as the final steps involved in the locking of loop D in a closed conformation. Finally, this new set of evidence pointing to serine 188 as essential for loop D gating regulation, coupled with the sequence variability existing at this position, highlight this residue as a putative key determinant of plant aquaporins’ functional diversity.

## 4. Materials and Methods

### 4.1. Molecular Dynamics Simulations

All simulations were performed with Gromacs (v.2018.1) [27] in a CHARMM36m force field [28]. The systems were built with CHARMM-GUI interface [29,30]. A first minimization step was followed by 6 equilibration steps, during which restraints applied on the protein backbone (BB: N CA C O), side chains (SC: side chains heavy atoms), and on lipids (LIPID: polar head heavy atoms) were progressively removed before the production phase was performed without restraint. Pressure and temperature were kept constant at 1 bar and 303.15 Kelvin (NPT ensemble), respectively, using the Berendsen method during equilibration and Parrinello–Rahman and Nosé–Hoover methods during production. The Lennard-Jones interaction threshold was set at 12 Angströms (Å) and the long-range electrostatic interactions were calculated through the particle mesh Ewald method [31]. The time step used for production is 2 fentoseconds.

#### Three Experimental Setups Were Carried Out

Experimental setup 1 corresponds to Figure 2 and Figure 3: the tetrameric assembly of SoPIP2;1 (pdb: 1z98 [16]) was inserted into POPC bilayer, and solvated with transferable intermolecular potential 3 (TIP3 [32]) water molecules and 150 mM of KCl. A phosphoryl group was added to serine 188 in each subunit to mimic its phosphorylation. The system was then simulated for 1.2 microseconds.

Experimental setup 2 corresponds to Figure 4: an open conformation was extracted from the previous experimental setup trajectory corresponding to chain D at time t = 871 nanoseconds. This conformation was chosen based on its high water permeability and because it corresponded to the widest opening of the channel intra-cellular mouth (see results). The subunit was then duplicated and aligned to the original crystal structure with PyMOL 2.4 software [24] to generate a new tetrameric assembly in a fully open state. This new tetramer was inserted into the POPC bilayer, solvated with TIP3 [32] water molecules, and 150 mM of KCl. The phosphorylation mark on serine 188 was kept. The system was then simulated for 100 nanoseconds.

Experimental setup 3 corresponds to Figure 5: similarly to experimental setup 2, an open conformation extracted from experimental setup 1 (time = 871 ns) was duplicated to form a tetrameric assembly. From this rebuilt open tetramer, three systems were obtained: _condition “open_S188Phos” where serine 188 was phosphorylated in all 4 sub-units; _condition “open” where serine 188 was kept without phosphorylation; and _condition “open_H193P” where serine 188 was kept without phosphorylation but histidine 193 was protonated on each sub-unit. These three tetrameric assemblies were inserted into the POPC bilayer, solvated with TIP3 [32] waters and 150 mM of KCl, and simulated for 500 nanoseconds each. A fourth condition corresponds to the first 500 nanoseconds of experimental setup 1 and was called “closed_S188Phos”.

### 4.2. Analysis

#### 4.2.1. Water Permeability

To monitor water molecules displacement along the trajectories, the MDAnalysis library was used [33,34]. From these water coordinates, water counts and permeability coefficients (pf) were derived. For water counts, a cylinder of 15 Angströms of radius and of 30 Angströms of length (which encompasses the whole transmembrane channel section of the AQP) was defined for each monomer and each frame. This cylinder was centered on the center of geometry of the alpha carbon of the two asparagine of the NPA motifs (Figure 1). One water count was defined when a water molecule crossed a cylinder from one extremity to the other. Permeability coefficients (pf) were calculated according to the collective coordinate method [35].

#### 4.2.2. Free Energy Profiles

Water free energy profiles were extrapolated from the logarithm function of the water counts inside the pore with the z-axis as a reaction coordinate [36,37]. The pore is divided along the reaction coordinate (z axis) in slices of 0.5 Angströms (Å). The average density of water molecules in each slice is then computed over the 200 nanoseconds portions of simulation and the Gibbs free energy G(z) is obtained as follows:
$$G\left(z\right)=-\mathrm{KT}\ln\frac{\rho (z)}{\rho_{\mathrm{bulk}}}$$
where K, ρ_bulk_, and T represent the Boltzmann constant, the bulk density, and the absolute temperature, respectively.

#### 4.2.3. Other Properties

Distances were computed with Gromacs tools (version 2020.6).

*Spicacia oleracea* PIP sequences retrieval and multiple alignment: to retrieve other PIP sequences, a Spinacia oleracea PIP2;1 amino acids sequence was blasted against non-redundant GenBank CDS database with default parameters. From the 35 sequences yielded this way, only 6 were annotated as PIP aquaporins and kept for multiple alignment. Multiple alignment was performed using MUSCLE web service [38] with default parameters and illustrated with Jalview [39] using the so-called Taylor color code. 

#### 4.2.4. Statistical Analysis

All statistical analyses were performed using R programming language [40] with the integrated development environment Rstudio. Before any statistical test was performed, the normality and homoscedasticity of the variables were controlled to choose between parametric or non-parametric tests. Thereafter, when two conditions were compared, either Student *t*-test or Mann–Whitney test was used and, when more than two conditions were compared, either Tukey’s post hoc test after one-way analysis of variance or Bonferroni post hoc correction after Wilcoxon test were used.

For experimental setup 1, the two conditions were extracted from the same trajectory based on water permeability and distances between loop D and N-terminal extension backbone alpha carbons to discriminate “closed” and “open” conditions. For each condition, the 200 nanoseconds portion of the trajectory was divided into 10-nanosecond sub-trajectories and the analyses were performed for chain D subunit, thus only yielding 20 repetitions per condition.

For experimental setup 3, the 500-nanosecond simulations of the four conditions were divided into 10-nanosecond sub-trajectories and the four monomers were taken into account, thus yielding 200 repetitions per condition.

## Figures and Tables

**Figure 1 ijms-24-13798-f001:**
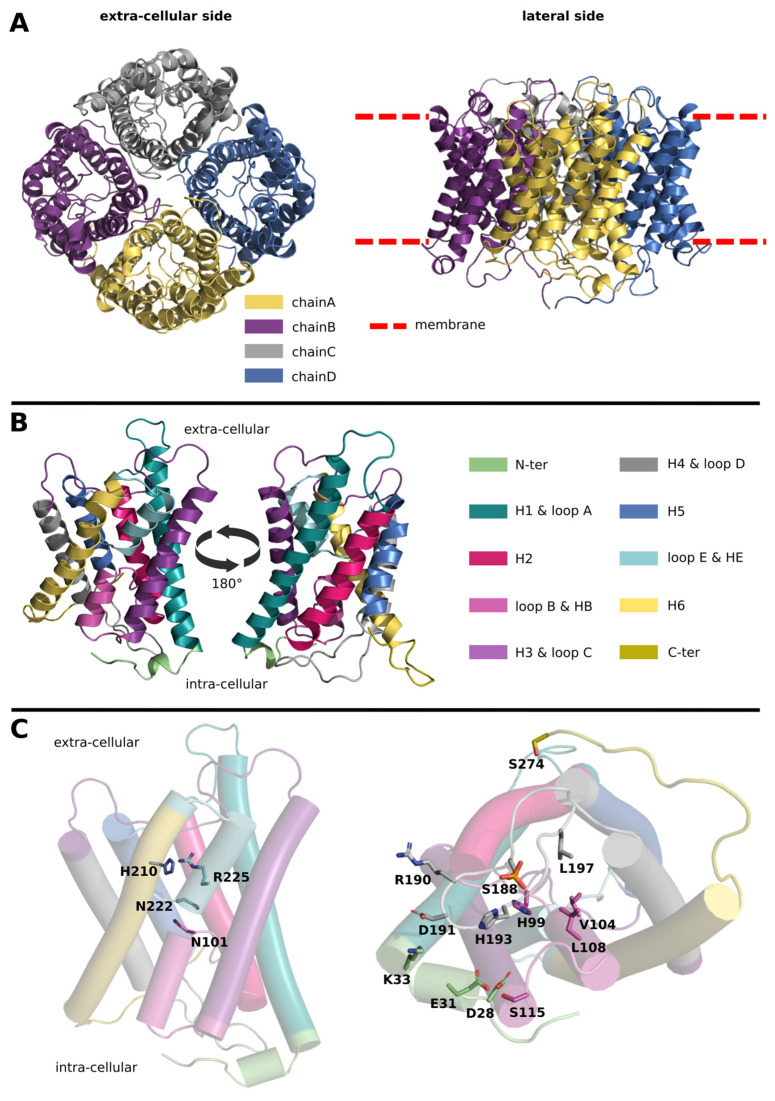
**The AQP fold and intracellular gating by loop D.** All the representations are made from spinach AQP SoPIP2;1 (pdb: 1z98). (**A**) Schematic representation of the AQP tetrameric assembly. Each subunit (or chain) is colored differently. (**B**) Schematic representation of the AQP fold. Each subunit is a functional water channel and is made of the following: N-terminal extremity (N-ter), helix 1 (H1), loop A, helix 2 (H2), loop B containing the small alpha helix HB, helix 3 (H3), loop C, helix 4 (H4), loop D, helix 5 (H5), loop E containing the small alpha helix HE, helix 6 (H6), and the C-ter. (**C**) Schematic representation of one subunit. On the left, two residues involved in the formation of the ar/R constriction, i.e., arginine 225 and histidine 210, as well as the two asparagines of the asparagine-proline-alanine (NPA) motifs which meet at the center of the channel, are represented. On the right, the channel is depicted as seen from the intracellular compartment in a closed state. All residues involved in plant AQP pH gating by loop D are represented.

**Figure 2 ijms-24-13798-f002:**
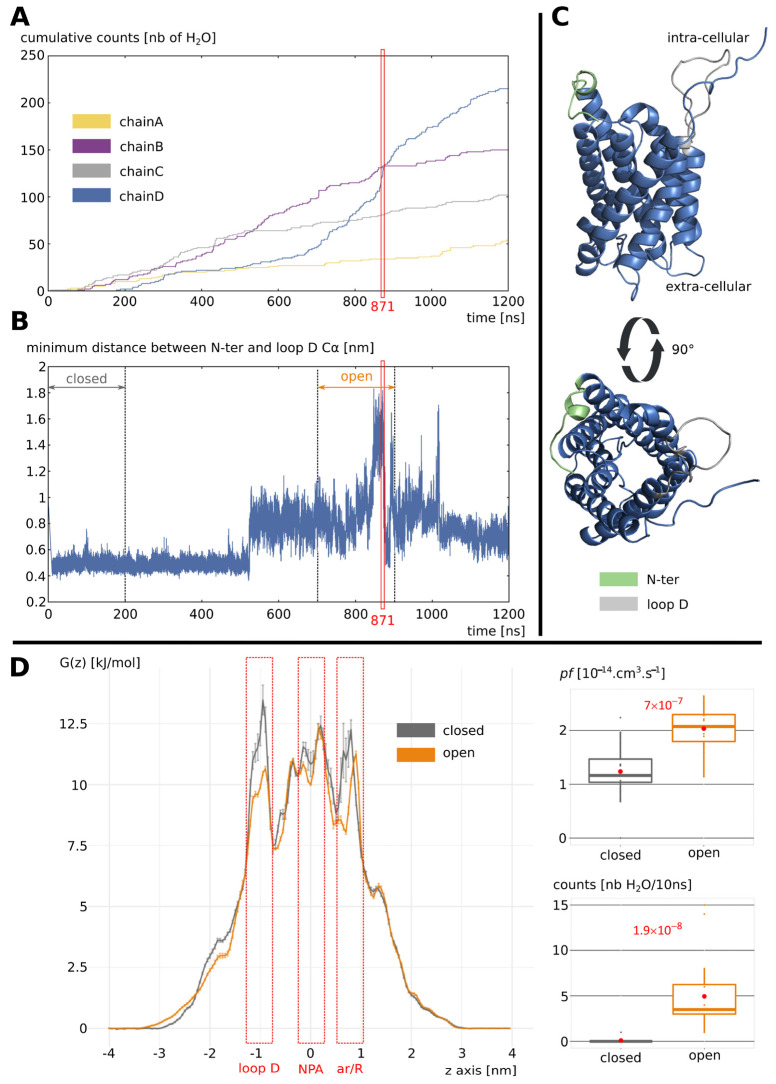
**Opening of the channel is triggered by phosphorylation of serine 188.** (**A**) Cumulative number of water molecules crossing the whole transmembrane 30 Angströms long channel section for each subunit. (**B**) Minimal distance between loop D and N-ter backbone alpha carbons (Cα). (**C**) Schematic representation of subunit chain D at time t = 871 ns. (**D**) Impact of the phosphorylation of serine 188 on the channel water permeability. On the left: free energy profiles of water molecules along the z axis and inside the channel of one subunit (chain D) in the beginning of the trajectory (t_start_ = 0 nanoseconds and t_end_ = 200 nanoseconds) in condition “closed” and in the end of the trajectory (t_start_ = 700 nanoseconds and t_end_ = 900 nanoseconds) in condition “open”. Three structural regions are indicated by red dashed boxes: the intracellular loop D; the central NPA motifs located on small helices HB and HE; and the aromatic/arginine constriction at the extracellular mouth of the channel. On the right: comparison of the two conditions with two permeability indicators. The permeability coefficient (pf) was compared using Student’s *t*-test and the counts using Mann–Whitney test. The corresponding *p* values are indicated in red.

**Figure 3 ijms-24-13798-f003:**
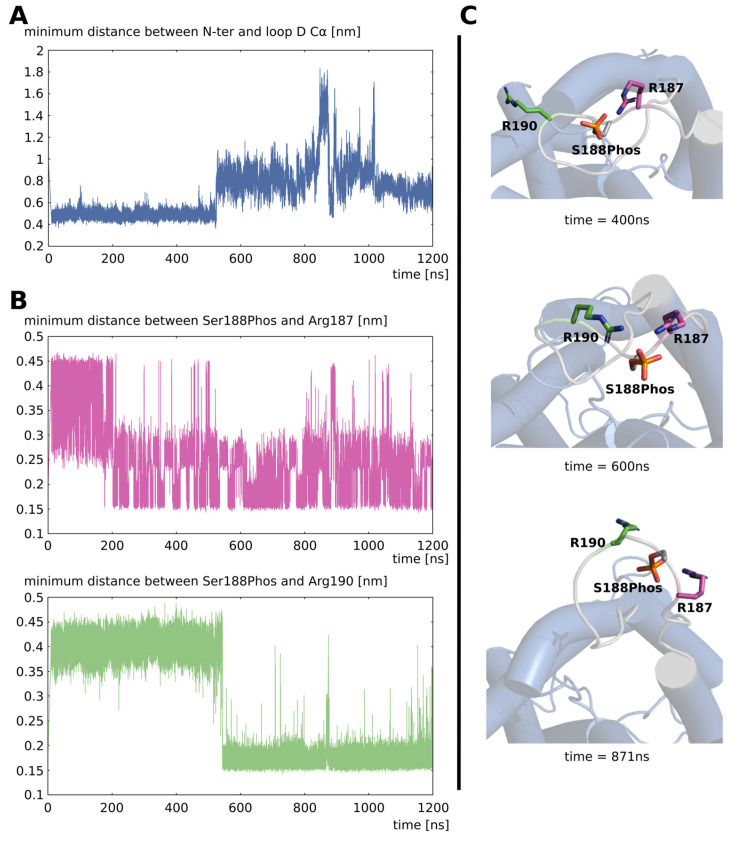
**The opening mechanism involves two arginines of loop D.** (**A**) Minimal distance between loop D and N-ter backbone Cα. (**B**) Minimal distance between phosphorylated serine 188 and arginine 187 or 190 side chains. (**C**) Schematic representation of chain D as seen from the extra-cellular compartment. Loop D is colored in gray and residues serine 188, arginine 187, and arginine 190 are represented.

**Figure 4 ijms-24-13798-f004:**
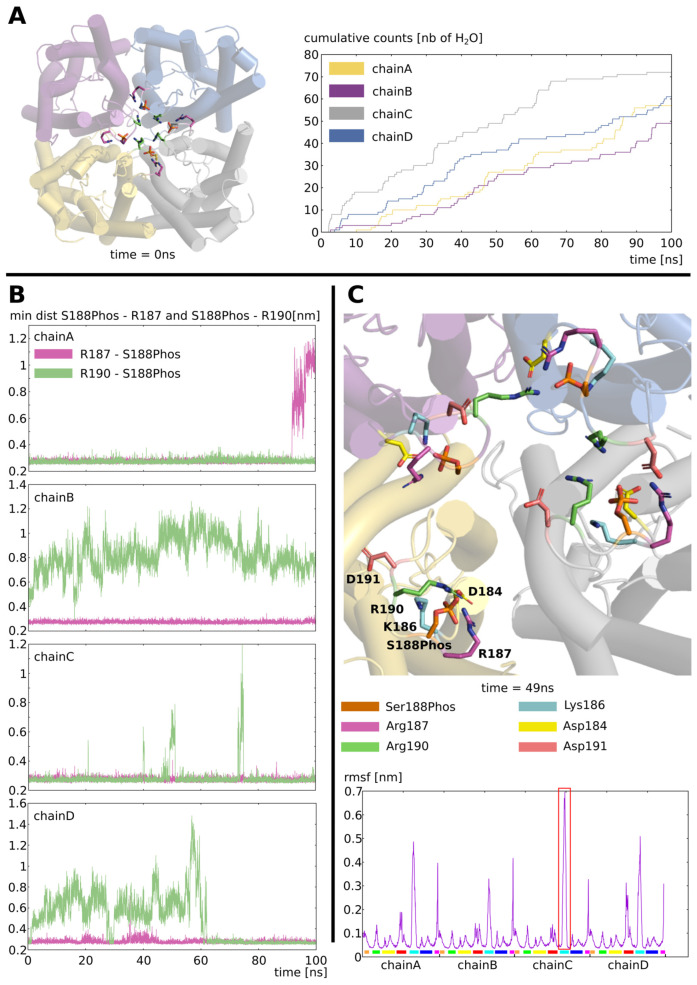
**Stabilization of the open state.** (**A**) On the left: schematic representation of the tetramer of SoPIP2;1 in a fully open conformation (see methods) as seen from the extracellular compartment at the beginning of the trajectory at t = 0 nanoseconds. The residues implicated in the opening of loop D are indicated: phosphorylated serine 188 in orange; arginine 187 in green; and arginine 190 in pink. On the right: Cumulative number of water molecules crossing the whole transmembrane section over the whole 100 nanoseconds simulated for each chain. (**B**) Minimal distance between serine 188 phosphoryl group and the guanidinium group of arginine 187 (in green) or arginine 190 (in pink) for each chain. (**C**) Schematic representation of the tetramer extracellular surface at mid trajectory (time t = 49 nanoseconds). The residues implicated in the opening of loop D and other charged residues of the same loop are indicated: phosphorylated serine 188 (S188Phos), arginine 187 (R187), arginine 190 (R190), lysine 186 (K186), aspartate 184 (D184), and aspartate 191 (D191). On the bottom part, root mean square fluctuation of SoPIP2;1 backbone atoms over the 100 nanoseconds of trajectory. The color code underneath corresponds to each flexible part of the protein: in orange—the N-terminal extremity, in green—the extra-cellular loop A, in yellow—the intracellular loop B, in red—the extracellular loop C, in cyan—the intracellular loop D, in dark blue—the extracellular loop E, and in pink—the C-terminal extremity. The part of the protein that fluctuated the most from the initial conformation corresponds to chain C loop D and is indicated by a red box.

**Figure 5 ijms-24-13798-f005:**
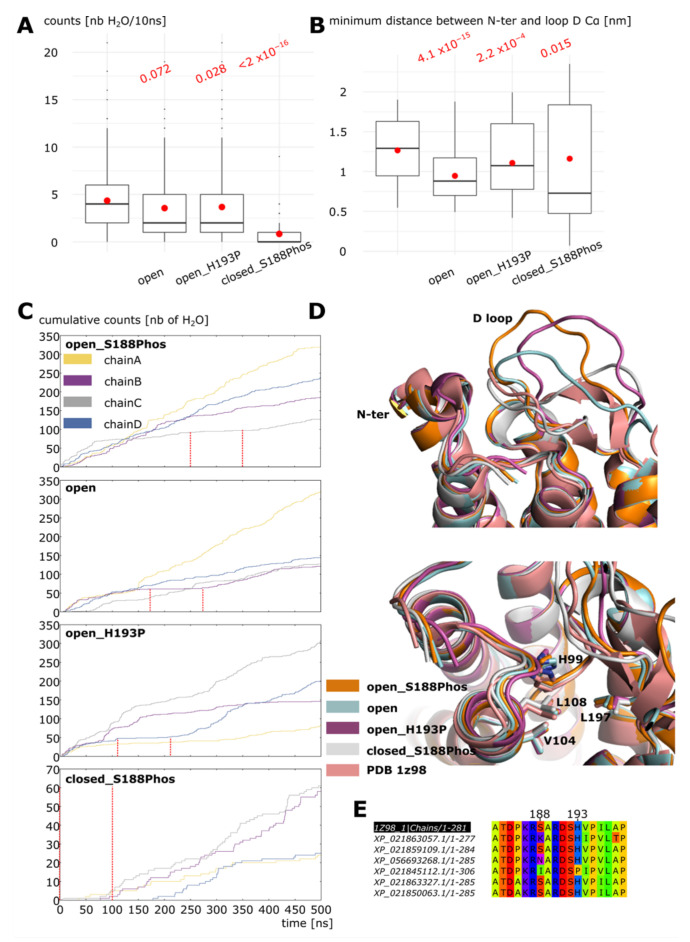
**Closing of D loop is triggered by dephosphorylation.** (**A**) Number of water molecules crossing the whole 30 Angströms long transmembrane pore section. All conditions were compared to one another with non-parametric Wilcoxon test. *p* values associated with comparisons of each condition with the “open_S188Phos” condition are indicated in red. (**B**) Minimal distance between the backbone atoms of N-terminal extremity and intra-cellular loop D. All conditions were compared to one another with non parametric Wilcoxon test. *p* values associated with comparisons of each condition with the “open_S188Phos” condition are indicated in red. (**C**) Cumulative number of water permeation for each of the 4 sub-units of the tetramer in each condition. The 100 nanoseconds portions of trajectory used for average structure alignment presented in (**D**) are indicated by red dashed lines: for condition “open_S188Phos”: chain C, time 250 ns to 350 ns; condition “open”: chain B, time 170 ns to 270 ns; condition “open_H193P”: chain D, time 110 ns to 210 ns; and condition “closed_S188Phos”: chain D, time 0 ns to 100 ns. (**D**) Structural alignment of average conformation of 100 nanoseconds portions of trajectory (see (**C**)) with experimental structure of the closed conformation (pdb 1z98). On the bottom part, side chains of residues histidine 99, valine 104, leucine 108, and leucine 197, which together form the hydrophobic occlusion characteristic of a closed pore, are represented. (**E**) Intracellular loop D portion of a multiple alignment of amino acid sequences of SoPIP2;1 with six other PIP from Spinacia oleracea. Position 188 and 193 of SoPIP2;1 are indicated.

## Data Availability

The data presented in this study are available on request from the corresponding author. The data are not publicly available due to the lack of a dedicated database.

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
