# Peer review of "Plant Aquaporin Gating Is Reversed by Phosphorylation on Intracellular Loop D—Evidence from Molecular Dynamics Simulations"

_ijms, 2023, doi:10.3390/ijms241813798_

Round 1

Reviewer 1 Report

The present study (“Plant Aquaporin’s gating is reversed by phosphorylation on 2 intra-cellular loop D. Evidences from molecular dynamics simulations”) submitted by Robin Mom, Stéphane Réty, Vincent Mocquet, and Daniel Auguin, describes an interesting topic: the dynamic gating of aquaporins modulated by post-translational modification such as phosphorylation. Understanding these types of issues in protein regulation are key to understand the functionality, although they are difficult to address in an in vivo or in vitro form. Therefore, employing molecular dynamics simulations could be a first step to delve deeper into the knowledge of aquaporin gating.  I believe that the paper provides relevant information in the area of study and would be potentially publishable in IJMS after addressing some aspects.

General comments:

·         Place all figures after the corresponding text to facilitate the reading of the manuscript.

·         Add one keyword corresponding to in silico analysis.

·         Remove all spaces before “ :”.

Abstract

The abstract lacks information, it is too short, so there is still room to extend it a bit. Most of the abstract focuses on the introduction, but the objective is not described, only what has been done, which is to investigate the mechanisms of plant aquaporin regulation. On the other hand, the results are simple and do not provide a final conclusion. The abstract can be schematised in a more standard structure with four parts: context, objective, method and results/conclusions (although without the need to write titles for each part).

The sentence "Plant aquaporins (AQP) are especially diverse." is too ambiguous, in which sense (sequence, functionality, family members….)?

Introduction

·         Line 33 – post-translational modification of AQPs are not cited as a mechanism of membrane water permeability, since post-translational modification interfere not only in gating but also in subcellular location and protein degradation.

·         Line 70 – It is not clear why the paper focuses on serine 188, as it is not named in the introduction, nor does it appear in the figure. Perhaps a hint should be given as to why it is thought to be relevant.

·         Figure 1 – I would try to place figure 1 in the introduction rather than in the results section.

·         The role and importance of phosphorylations in proteins are not introduced.

Results

·         Figure 2 – I would try to place figure 2 after the corresponding section (2.1).

·         Line 81 – the citation of figure 2 here seems to be inaccurate, since figure 2 not represent “serine 188 in a phosphorylated state in each subunit”.

·         Figure 2A – indicate in figure what time correspond to open conformation

·         Line 107 – “This choice was based on both the number of water molecules crossing the 107 whole transmembrane section of the channel and the mean minimal distance between 108 loop D and N terminal residues backbone” looks like discussion and not like results.

·         Line 109 – the one observed in the figure for loop D has been checked for the other loops?

·         Since in section 2.1. The NPA region and the aromatic/arginine constriction are used in the explanation of the results, it may be interesting to show them in Figure 1.

·         Line 128 – remove space after “follow” (follow :).

·         Line 170 – define what rmsf is.  

·         Line 179 – remove space after “5) “.

·         Figure 5 – Why are the statistics shown in figures 5A and 5B different? If you want to do multiple comparisons why not an ANOVA, or at least show the statistics in both figures in the same way?

·         What did you expect to find when you put the open_H193P conformation into the study?

·         Figure 5.E. - The purpose of multiple alignment of PIP sequences is not clearly explained in the manuscript.

Discussion

·         Line 261 – change "4" to "four sequences".

·         Line 309 – the conclusion corresponding to histidine 193 seem to be not relevant if the objective of the study is taking into account.

Materials and methods

·         Line 326 – reference of Ewald method ?

·         Line 329 – define “TIP3”

·         Line 353 – extend information about water permeability analysis.

·         Line 376 – add reference of R and if R Studio has been used.

References

·         Reference 1 is missing the journal title.

·         Reference 15 is missing the journal title.

·         References 18 and 19 indicate that are doctoral thesis.

·         Reference 23 is missing the journal title.

·         In reference 24 the journal title appear in the abbreviate form (Nat Methods)

·         Reference 26 is missing the journal title.

·         Reference 31 is missing the doi.

Author Response

 Dear reviewer,

First of all, on behalf of all the co-authors, I would like to thank you for the time and energy you dedicated to the reviewing of our work.

About your comments:

Abstract:

The abstract has been reformulated.

Introduction:

All of your comments have been taken into account. I believe the introduction is now more comprehensible, thanks to your critical reviewing.

Results:

The mistakes you have pointed out have been corrected. Figure 1 has been modified in order to display NPA motifs and ar/R constriction.
The figure 2 has not been modified however. Since each water count (which are already indicated on the figure by the curves) implies that the channel was open (at least transiently). The water molecules start crossing the channel arount time = 100 nanoseconds which hence corresponds to the beginning of the cumulative water counts curves (figure 2.A). However the D-loop reached a fully open conformation around time = 700 nanoseconds for the sub unit that we used for further characterization (chain D).
About the following sentence: “This choice was based on both the number of water molecules crossing the 107 whole transmembrane section of the channel and the mean minimal distance between 108 loop D and N terminal residues backbone”. The point was to compare functional subunit with non-functional subunit hence we used this two criteria to be sure of the functional status of the monomer.

All statistics were performed using the same procedure (described in methods). The conditions were compared with a non-parametric equivalent of ANOVA. The figure5 was modified to display p.values in the same way. Crystallographic structure of the closed state has also been added to structural alignment (figure 5.D).

The text has been modified to clarify the role of H193 protonation and to better explain the multiple alignment.

Discussion:

The text has been modified as well.

Methods:

All your comments have been taken into account.

References:

All your comments have been taken into account.

As a conclusive note, I appreciate the relevant comments you provided to us which I believe helped me to significantly meliorate the quality of our work.

Kind regards,

Robin Mom on behalf of all the co-authors

Reviewer 2 Report

The manuscript "Plant Aquaporin’s gating is reversed by phosphorylation on intra-cellular loop D. Evidences from molecular dynamics simulations." uses molecular dynamics simulations to investigate the gating mechanism of plant aquaporins. The manuscript is well-written and the findings contribute valuable insights into aquaporin regulation. However, I do have some suggestions to improve the clarity and impact of the work:

Major comments:

1.     The introduction provides good background on aquaporin gating and the current knowledge gaps, but it would benefit from more clearly laying out the specific objectives and hypotheses tested in this study. What exactly is novel about looking at phosphorylation of Ser188 as a trigger for opening?

2.     More details should be provided on the simulation parameters, force fields, etc. in the Methods section. For example, what was the length of each simulation, time step used, ensemble, treatment of long-range electrostatics, etc.? This is important for reproducibility.

3.     The authors should discuss in more detail how their findings compare with previous computational and experimental studies on aquaporin gating. For instance, how do their results on Ser188 phosphorylation agree or contrast with the past work cited in the introduction? Some added discussion of convergence with other approaches would strengthen the study.

4.     The manuscript would be strengthened by some analysis of the interactions and hydrogen bonding networks involved in stabilizing the open and closed states. Were there any striking differences between the simulations and crystal structures in this regard?

5.     Unify the style of the references in the References Section and add DOI in the cases it is possible. And use the same reference and citation (follow MDPI’s guidelines) style in the main text.

Minor comments:

1.     The quality of Figure 2D could be improved; the text is difficult to read.

2.     There are some typos throughout that need correcting (e.g. line 276 "agnströms")

3.     The authors need to define any abbreviations on first use (e.g. MD)

Author Response

Dear reviewer,

First of all, on behalf of all the co-authors, I would like to thank you for the time and energy you dedicated to the reviewing of our work.

Major comments :

  1. The introduction has been extended, with more details about the reasons why we chose to investigate the phosphorylation mark of serine 188 further.

  2. All the mentioned details have been specified in the methods section.

  3. We have discussed our results with other approaches. One study in particular was used as a starting point and a reference for comparison : the Nyblom et al study where the authors evaluated the impact of S188 phosphorylation through Xenopus laevis oocytes measurements, crystallography and molecular dynamics.

  4. The open and closed conformations have been characterized through the minimal distance between D-loop and N-ter and through water permeability measurements. For the open conformation, we observed that the D-loop was oscillating between several open conformations because of the dynamic formation and rupture of salt bridges with the phosphoryl group. And for the closed conformations, figure 5.D presents a structural alignment with the crystallographic structure. The D-loop never reached the same conformation however, key residues were positioned in the same way. All of these points are described with more details in the results and discussions sections.

  5. The references have been unified.

Minor comments :

  1. All figures have been made with the same text size and quality.

  2. All typos have been checked out

  3. All abbreviations have been defined

Kind regards,

Robin Mom on behalf of all the co-authors

Round 2

Reviewer 2 Report

The authors have responded satisfactorily to the reviewer's comments and suggestions.